# Cavitand Decorated Silica as a Selective Preconcentrator for BTEX Sensing in Air

**DOI:** 10.3390/nano12132204

**Published:** 2022-06-27

**Authors:** Andrea Rozzi, Alessandro Pedrini, Roberta Pinalli, Enrico Cozzani, Ivan Elmi, Stefano Zampolli, Enrico Dalcanale

**Affiliations:** 1Department of Chemistry, Life Sciences and Environmental Sustainability and INSTM UdR Parma, University of Parma, Parco Area delle Scienze 17/A, 43124 Parma, Italy; andrea.rozzi@unipr.it (A.R.); alessandro.pedrini@unipr.it (A.P.); roberta.pinalli@unipr.it (R.P.); 2PROAMBIENTE S.c.r.l., Via P. Gobetti 101, 40129 Bologna, Italy; e.cozzani@consorzioproambiente.it; 3CNR-IMM Bologna, Via P. Gobetti 101, 40129 Bologna, Italy; elmi@bo.imm.cnr.it (I.E.); zampolli@bo.imm.cnr.it (S.Z.)

**Keywords:** preconcentrator, quinoxaline cavitand, BTEX sensing, air monitoring, MEMS column

## Abstract

The monitoring of benzene and other carcinogenic aromatic volatile compounds at the ppb level requires boosting both the selectivity and sensitivity of the corresponding sensors. A workable solution is the introduction in the devices of preconcentrator units containing molecular receptors. In particular, quinoxaline cavitands (QxCav) resulted in very efficient preconcentrator materials for the BTEX in air to the point that they have been successfully implemented in a commercial sensor. In this work, we report a highly efficient quinoxaline-based preconcentrator material, in which the intrinsic adsorption capacity of the QxCav has been maximized. The new material consists of silica particles covalently coated with a suitable functionalized QxCav derivative (QxCav@SiO_2_). In this way, all the cavities are exposed to the analyte flux, boosting the performance of the resulting preconcentration cartridge well above that of the pure QxCav. It is noteworthy that the preconcentrator adsorption capacity is independent of the relative humidity of the incoming air.

## 1. Introduction

Environmental monitoring represents one of the greatest challenges for the chemical community due to the stringent sensor requirements in terms of sensitivity and selectivity. Preconcentrators have found wide acceptance as an effective solution to impart both sensitivity and selectivity to sensor devices [1]. Their use is particularly appealing for the real-time detection of benzene, the carcinogenic component [2] of benzene, toluene, ethylbenzene and xylenes (BTEX) at low ppb levels in air [3]. The issue of achieving molecular-level selectivity and low-ppb sensitivity at the same time can be addressed by disconnecting the recognition element from the detection unit by using molecular receptors as preconcentrators. In this context, our group has pioneered the use of cavitands as selective preconcentrators in indoor and outdoor sensors for BTEX [4]. The outcome of this activity is a commercially available stand-alone device [5] capable of monitoring the concentration of benzene in air below the current EU limit value of 5 µg m^−3^ [6]. In its most recent configuration, the device is composed of a micro-electro-mechanical system (MEMS) cartridge packed with a solid cavitand receptor to selectively trap the BTEX during sampling at room temperature. The separation of the different aromatic compounds released by the cavitand preconcentrator after thermal desorption is obtained by interfacing the preconcentrator with a GC-MEMS column [7]. The separated analytes were then individually channeled to a photoionization detector (PID), achieving a detection limit in the sub-ppb range.

The tetraquinoxaline cavitand QxCav (Figure 1) is a molecular receptor featuring a preorganized cavity of nanosize dimensions [4]. It has been shown to be selective for the uptake of aromatic analytes in air in the presence of larger amounts of aliphatic hydrocarbons [8,9]. Moreover, QxCav is insensitive to air humidity and to the inorganic pollutants present in air such as CO_2_, NO_x_ and SO_x_. Its remarkable selectivity at the gas–solid interface is attributed to CH–π interactions between the analytes and the inner cavity aromatic surfaces [10,11]. In the present device configuration, QxCav is loaded in the MEMS cartridge as amorphous powder with particles 180–250 µm in size [7]. This particle size distribution has been optimized to allow optimal air flux through the preconcentrator and to be easily retained by an MEMS mechanical filter. This solution has the advantage of synthetic simplicity, but it brings in some undesired issues: (1) only cavitands on the particles surface are active in BTEX uptake as most of them are buried in the bulk solid; (2) cavitands are randomly oriented, and only those with the mouth of the cavity exposed can trap the analytes; and (3) the cavitand particles tend to become smaller and smaller by mechanical stress due to flow pumping while cycling at high temperature, causing a progressive increase in MEMS impedance and a consequent loss of performance. These issues prompted us to consider the grafting of QxCav on solid supports as a workable solution [12,13].

Here, we report a structurally designed, highly efficient preconcentrator material for BTEX adsorption, in which the QxCav is covalently coated on the surface of silica particles. The resulting material overcomes the two undesired issues reported above related to the use of pure cavitand powder as a preconcentrator. The new preconcentrator is used in the final device to test its performances under real working conditions in comparison with pure cavitand particles and bare silica.

## 2. Materials and Methods

The experimental details, including the materials and apparatus, are listed in the Appendix A. The precursor QxCav with four ω-decenyl feet was prepared by following a previously published procedure [14]. Mono-hydrosilylated QxCav was synthetized by adapting a previously published procedure [15].

### 2.1. Monosilylated QxCav

300 mg (194 µmol, 1 eq) of QxCav with four ω-decenyl feet was dissolved in 30 mL of dry toluene. Four freeze-pump-thaw cycles were performed to degas the solution, and 226 µL of triethoxysilane (95%, 116 mmol, 6 eq) and 75 µL of Karstedt’s catalyst (xylene solution, Pt~2%) were added under nitrogen atmosphere. The mixture was stirred for 48 h at room temperature. Then, the reaction was quenched into 250 mL of methanol and the white solid was filtered and washed with methanol. The product became pale brown after drying (235 mg, 65% yield) (see Appendix A for characterization).

### 2.2. Silica Grafting Procedure

500 mg of silica (180–250 µm size) were placed in a two-necked round-bottom flask and dried by heating under vacuum. Monosilylated QxCav (250 mg) was dissolved in the minimum amount of dry toluene (15 mL) and added to the reaction flask. The suspension was kept at 100 °C overnight under gentle stirring. After cooling to 25 °C, the solid was filtered on a sintered glass. The solid residue was abundantly washed first with dichloromethane and then with methanol to remove the unreacted cavitand. The functionalized silica appeared to be brownish, and the weight percentage of the organic layer deposited was quantified as 15% by performing TGA from 30 to 900 °C at 10 °C/min in air (Figure 1).

### 2.3. Preconcentrator Testing Protocol

The QxCav@SiO_2_ preconcentrator was tested in the commercial mini-GC system PyxisGC BTEX, equipped with a MiniPID 2 ppb by ION Science (UK) [5]. The same measurements were conducted on bare silica and pure QxCav as control experiments. Each MEMS cartridge (Appendix A) was filled with 50 mm^3^ of bare silica, QxCav and QxCav@SiO_2_, respectively, corresponding to roughly 100 mg of material for silica and QxCav@SiO_2_ and 50 mg of QxCav. The sampled air contained a mixture of BTEX from a certified cylinder, each of them at a 3-ppb concentration, except for the p-xylene present at a 6-ppb concentration. The air was humidified at 30, 50 and 80% RH to mimic real ambient conditions. A flow of 250 sccm of the humid sampled air was pumped into the preconcentrator for 300 s at 25 °C. At the end of preconcentration, the embedded 3-way mini-valve was switched, connecting the preconcentration unit to the GC pump. The temperature of the MEMS was raised up to 110 °C to promote the desorption of the analytes with an inverted air flux of clean indoor air. The desorbed analytes were injected into the MEMS GC separation column for 40 s. The mini-GC column operated in a temperature ramp from 50 to 185 °C in 140 s in order to maximize the separation of peaks. A typical chromatogram is reported in Figure 2. The detection of each analyte was provided by a photo ionization detector (PID) positioned at the end of the MEMS column. At the end of injection, the preconcentrator was cleaned for 2 min at 110 °C. After cooling the entire apparatus to 25 °C in 120 s, new air sampling began. A single measurement takes 10 min to be completed, enabling a benzene detection limit of 0.8 µg m^−3^. Longer cycle times allowed for even higher sensitivity by increasing the sampling time (Figure 3a). As shown in Figure 2, all analytes were separated well under these experimental conditions.

The elution time was constant for every analysis for all analytes. At least 20 measurements were collected for each combination of experimental parameters and stationary phases. The resulting histograms reported in Figure 3 are the averages of 20 measurements, comparing the heights of the GC peaks for each preconcentration material tested.

## 3. Results and Discussion

### 3.1. Preparation and Characterization of the Preconcentrator Material

For the covalent grafting of QxCav on the silica surface, monosilylated QxCav was prepared by adapting previously reported procedures [15,16]. Karstedt’s catalyzed hydrosilylation between the ω-decenyl-footed QxCav and an excess of triethoxysilane afforded a mixture of differently functionalized cavitands (Figure 2). Its composition was assessed by 1D and 2D nuclear magnetic resonance spectroscopy (NMR, Appendix A), matrix-assisted laser desorption/ionization time-of-flight mass spectrometry (MALDI-TOF, Appendix A) and FT-IR spectroscopy (Appendix A). Although no residual signals ascribable to a terminal double bond were observed in ^1^H NMR, only mono- and di-functionalized products were observed in the mass spectrum as a consequence of olefin isomerization side reaction in the hydrosilylation protocol. An average of 1.3 silyl groups per cavitand was estimated by ^1^H NMR signal integration. Surface decoration was performed by reacting the as-obtained mixture in boiling toluene with pre-meshed silica (180–250 µm, Figure 2). The ratio between the cavitand and silica particles was 1:2 *w*/*w*. The weight percentage of the deposited organic layer was quantified to be 15% by TGA (Figure 1). The initial 2% loss observed in the TGA thermogram was ascribable to the release of the cavity-included and surface physisorbed toluene, which was used as a solvent in the silica grafting procedure.

### 3.2. Uptake and Release Capacity of the Material

The adsorption performances of the three preconcentration materials, namely bare silica, pure QxCav and functionalized silica (QxCav@SiO_2_), are summarized in the histograms of Figure 3. Each histogram reports the analyses performed at different percentages of relative humidity (RH). In all cases, the QxCav@SiO_2_ outmatched both the bare silica and pure QxCav in BTEX adsorption by far. The formation of a QxCav monolayer on the silica, with all cavities available for complexation, increased the preconcentrator uptake ability dramatically independent (within the error range) of the relative humidity of the incoming air. This is in sharp contrast with the behavior of the bare silica, whose BTEX physisorption was highly dependent on the relative humidity. The reduced silica retention at a high RH could be attributed to its hydrophilicity; water competes with organic molecules for surface physisorption. Despite the much larger amount of loaded cavitand (50 mg of pure QxCav versus ≈15 mg of QxCav coating the silica, based on TGA analysis), the QxCav preconcentrator was able to adsorb a much lower amount of BTEX. In the QxCav powder, the number of accessible cavities at the air–solid interface was limited, and the cavity orientation at the interface was random. Its RH response was similar to that of QxCav@SiO_2_ since they both presented hydrophobic cavities.

It is also worth noticing the cavitand bias toward benzene and toluene with respect to ethylbenzene and p-xylene, as evidenced by the chromatogram in Figure 2 and by the histograms in Figure 3b–d [17]. This selectivity was the result of the interaction mode of the aromatic hydrocarbons with the cavity, which was mainly driven by CH–π interactions between the guests’ hydrogens and the cavity’s aromatic walls. Two of them were with the lower part of the cavity, as demonstrated by the molecular mechanic calculations based on the cavitand complexes’ crystal structures [18]. Therefore, the presence of two methyl groups, as in p-xylene, destabilized the complexation. On the other hand, substituents larger than methyl, as in the case of ethylbenzene, introduced steric hindrance at the mouth of the cavity, thus jeopardizing the complexation. The preferential inclusion of B and T was pivotal for quantitative detection of carcinogenic benzene in air.

Regarding the particle size shrinking, no impedance increase was detected during the entire QxCav@SiO_2_ preconcentrator set of experiments, thus supporting the dimensional stability of the coated silica particles under service conditions.

## 4. Conclusions

In summary, grafting the QxCav onto precisely sized silica particles via hydrosilylation allowed the full exposure of the receptor cavities to the analyte flow. The remarkable selectivity exhibited by the QxCav cavity allowed the preferential uptake of benzene and toluene in the presence of overwhelming amounts of water in air. The combined effect of complete cavity exposure to the gas phase and its intrinsic selectivity led to a remarkable increase in BTEX uptake. The resulting QxCav@SiO_2_ preconcentrator showed superior properties in benzene detection with respect to the preconcentrator filled with the pure QxCav powder and the unfunctionalized silica particles. These unique properties of QxCav@SiO_2_ make it the preconcentrator of choice for benzene sensing in outdoor and indoor air. Furthermore, this miniaturized MEMS preconcentration unit is amenable to be interfaced with other transducers, like metal oxide semiconductors (MOSs) [19] or ion mobility spectrometers (IMSs) [20,21].

## Data Availability

Not applicable.

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
