# Peer review of "Cavitand Decorated Silica as a Selective Preconcentrator for BTEX Sensing in Air"

_nanomaterials, 2022, doi:10.3390/nano12132204_

Round 1

Reviewer 1 Report

The authors present a paper to boost both selectivity and sensitivity of monitoring system measuring the BTEX in air. The authors proposed a based cavitand decorated silica concentrator to achieve this goal. This paper is very interesting in my opinion but the document needs revisions.

-       Line 28-29 “Their use is particularly appealing for the real-time detection of benzene, the carcinogenic component of BTEX, at low ppb levels in air”… The authors should add other references to strongly reinforce the terms “ carcinogenic component”

-       Line 35-36 “capable of monitoring the concentration of benzene in air below the current EU limit value of 5 µg m-3 “. The authors should add some information about the measurement  sampling time to achieve a valid data or add the measurement frequency that can be used (e.g. one measurement at minutes, 10 minutes or few seconds, …). This is important to clarify the term as “real time measurements”.

-       Line 34-65 “In the present sensor configuration, QxCav is loaded in the MEMS cartridge as amorphous powder with particles of 180–250 µm size [6]” ….the author should better rearrange the text and the conceptual sense here reported. In fact in my opinion, speaking in the same paragraph of  “in the most recent configuration of commercially available stand-alone sensor system developed by  authors in the past….“ and then “ ….speaking about the pre-concentrator with a new method of developing...” this confuses the reader. In fact I have not understand if there is a sensor that is based on cavitand in the  commercial one or a simple pre-concentrator with a PID as sensor.

-       Please add a paragraph that report the novelty in clear way.

-       Line 46 “Moreover, it is totally insensitive to air humidity…..”. in my opinion if the cavitand could be insensitive to humidity ….the silica or the support (or scaffold) material it is not. Please add some comments about this or use other terms instead of “totally”.

-       Line 72-87. Did the authors check the presence of residual solvent (e.g. Toluene, Xylene) after the production .? and then if it was present, how they totally have removed it ?

-       The authors, in the paragraph 2.2 should add a figure that reports the same measurement performed with a standard CGMS to compare the performances.

-       Line 105 The authors should provide some information about the PID that has used.

-       In figure 2 the authors report xylene as label of peak. Is it correct ? please change it if it was p-xylene or other xylene isomers.

-       in my opinion the title of paragraph 3.2 should be changed, since the authors cannot speak of sensing but only of “adsorption or absorption or uptake  capacity of the material”. Please add comments on this.

Author Response

See enclosed file

Reviewer 2 Report

The Enrico Dalcanale et al. represents synthesis of novel cavitand-based preconcentrator for BTEX sensing in air. Resorcinarene macrocycle functionalized by quinoxaline groups on the upper rim and long chained hydrocarbon tails on the lower rim were incorporated to the structure of silica microparticles to give novel hybrid nanomaterial capable of highly selective BTEX sensing. The paper contains good level of originality and novelty. A numerous set of physical methods is used in this work (mainly in Supplementary) strengthening the overall quality of the paper and supporting suggested conclusions. However, following issues should be thoroughly addressed in order to deliver more clarity and to reveal the details for the broad readership of the Nanomaterials.

1.      Acronym BTEX should be decrypted at the first mention or definition should be given.

2.      There is no information about silica surface functionalization in introduction section, which is essential to unveil in this paper. Thus, it should definitely be strengthened by few examples of silica microparticles’ surface decoration, including very relevant 10.3390/ijms20133139. Moreover, the authors are encouraged to refer following recent review (10.3390/molecules26051214) to support the role of cavitands for different applications.

3.      The first step of weight loss on the Figure 1 assigned to physisorbed toluene should be proved by GSMS as well as second wave.

4.      1:2 cavitand-SiO2 ratio seems too excessive for organic component towards bulky and voluminous (180-250 µm size) SiO2 particles. It is necessary to analyze supernatant after filtration for the reason of presence of unreacted QxCav.

5.      Please provide TEM images of functionalized and non- functionalized SiO2 particles.

6.      What was the specific reason of pretty large SiO2 microparticles choice for this study?

7.      The selectivity within the row of BTEX is good. What is the effect of typical interfering reagents on selectivity?

Author Response

See enclosed file

Round 2

Reviewer 1 Report

it is ok in the present form.

Reviewer 2 Report

The reviewer thanks the authors for addressing all comments. The manuscript can be recommended for publication in Nanomaterials.